# OpenReview forum: "Moral Alignment for LLM Agents"
_ICLR.cc/2025/Conference — ICLR 2025 Poster_

### Official Review · Reviewer_yMDj · 2024-10-29

**Soundness:** 3
**Presentation:** 3
**Contribution:** 2
**Rating:** 6
**Confidence:** 2

**Summary:**

This paper explores the possibility of aligning LLM agents by fine-tuning on intrinsic  moral rewards. Instead of implicitly representing human values during fine-tuning, as is done in methods like RLHF/DPO, the authors suggest directly specifying alignment goals by defining rewards based off moral philosophy frameworks. Namely, they fine-tune LLMs on the Iterated Prisoner’s Dilemma game using Deontological and Utilitarian rewards. They evaluate their approach using repeated social dilemma games in two settings: LLM vs another LLM, and LLM vs a tit-for-tat opponent. The authors then discuss how this can be generalized to other matrix games.

**Strengths:**

1. Aligning LLM agents to human values is a highly relevant and important research area.
2. The idea is interesting and I have no issues with any of the experiments. The authors do a good job of identifying the limitations of their method themselves, and other than those this is well-written research. Experimental results are thorough and comprehensive, and well cover key concerns I had as I read (e.g. anonymized names of action tokens, randomizing the action token order, etc). The paper is well-written and motivates their approach clearly.

**Weaknesses:**

As mentioned above, my main concerns here have been pointed out by the authors themselves already. Namely:

1. “A limitation of this approach
is that it requires the specification of rewards for a particular environment”, and
2. “An extension of this method to other environments would be of great interest, including fine-tuning agents with other payoff structures, more complex games or longer history lengths”

My primary concern is point 2; in my opinion, as-is this is a slightly weak paper and I would love to see this idea extended to more complex and interesting settings. Due to these limitations, I don’t believe this to be super impactful work - but I do think it is an interesting start. Hence, I would say it is marginally above the acceptance threshold.

**Questions:**

n/a

---

> ### Author Response · Authors · 2024-11-25
> **Response to Reviewer yMDj**
>
> We thank **Reviewer yMDj** for their comments. We would like to respond to the following comment in particular: *“I would love to see this idea extended to more complex and interesting settings”*. We must highlight the tradeoff that has to be made between the depth in which we can analyze the impacts of moral fine-tuning and the range of scenarios we are able to use in any one paper. To extend to more complex scenarios, it is important to first examine the simple and classic, widely-used 2x2 matrix games in an interpretable and systematic way. To the best of our knowledge, this is the first study that focuses on agent fine-tuning with intrinsic rewards for moral alignment. Also for this reason, we prioritized depth of analysis over breadth. We would like to quote **Reviewer kTWD** (who said our paper addresses *“a critical and highly understudied problem with a clever and interesting solution”*) in highlighting the potential impact of this work even at this early stage, as it introduces a novel methodology in the crucial field of AI Alignment.
>
> Generalization is a broader problem in Reinforcement Learning-based systems (and AI/ML systems in general). As a slight addition towards this goal, we would like to refer the Reviewer to our extended analysis of generalization beyond matrix games, where we have added evaluations of two further prompt formats representing the IPD game, as suggested by **Reviewer Jn1h** (see Figure 17 in the paper and the discussion above). These results point to the fact that the policies learnt in our structured setup may indeed be able to generalize to similar decision-making settings represented in less structured ways at test time.
>
> To extend this insight, four research agenda includes more complex scenarios and reward functions, including games with more than two actions, sequential games, and spatial dilemmas. We consider this work a fundamental methodological starting point of this investigation. We are aware that this is a somewhat concise discussion and this is an area of great interest for the community for the coming months and years.

---

> > ### Comment · Reviewer_yMDj · 2024-11-27
> >
> > Thank you for the response, and for what is clearly a very large amount of effort that has been put into these rebuttals. I agree one single paper cannot provide both sufficient depth and a whole field's worth of breadth - of course. It is perfectly fine to lay just the very first groundwork steps in an important direction, but there does need to be at least some sense the road that lies past that step is plausible and navigable. There certainly is space in one paper to highlight more details of what a path forward could look like. Overall, I believe my initial judgement to be fair: a well-written paper on a topic that needs more research, but a little weak and not doing as much as I'd like to really be laying groundwork for future research.

---

### Official Review · Reviewer_Jn1h · 2024-11-02

**Soundness:** 4
**Presentation:** 3
**Contribution:** 1
**Rating:** 5
**Confidence:** 4

**Summary:**

The paper introduces a new method for moral alignment of LLMs. Instead of using human preference comparisons, they suggest using intrinsic rewards to train an LLM for moral behavior. Specifically, they train on a dataset of games that are strategically equivalent to an iterated prisoner's dilemma, using two types of intrinstic reward: utilitarian and deontological. When trained with deontological rewards, models learn to cooperate against cooperators and defect against defectors, even when they were previously trained using game rewards and ended up always defecting. The paper investigates both training against a static opponent (tit for tat) and self-play. They test how the trained model generalizes to other general-sum bimatrix games and find that the models mostly continue to follow the moral framework they have been trained on.

**Strengths:**

- LLM moral alignment is an important topic, and especially the study of LLM agents in strategic interactions.
- I was able to follow along with the experimental methodology and results and they seem sound to me.
- Defining rewards intrinsic to games and training on these is an interesting idea that could help with alignment and avoid problems with inadequacies of human feedback. Moreover, game rewards might scale better than collecting human data.
- The authors do a great job visualizing the models' strategies graphically.
- The authors test generalization by using different tokens for the actions in the test games, to avoid simple token matching when generalizing.
- The authors compare extensively to relevant baselines, such as the base model, the game payoffs, and different ablations. They also perform both self-play and static opponent experiments. In general, I really appreciate all the thorough ablations.
- The authors use implicit ipd descriptions rather than specifically mentioning ipd/cooperate/defect, to avoid confounders from the model's preexisting knowledge about the ipd.

**Weaknesses:**

- As far as I can tell, the deontological rewards are hand-designed. As such I don't currently see how this approach could be scaled to more games with more complex action spaces. Moreover, in the specific ipd setting, deontological rewards basically amount to specifically training the LLM to implement a tit-for-tat policy. The fact that the model can learn this policy when specifically trained on hand crafted rewards is not that interesting.
- When it comes to utilitarian rewards, these are not hand-designed, but I feel like training on utilitarian rewards is not interesting. The point of a strategic situation like the prisoner's dilemma is to model real-world conflict scenarios. Of course, if agents are all cooperative and care about each other, one can avoid conflict. But this is not feasible in the real world, where LLM agents need to act in the interest of the user. They should still be able to cooperate, but LLM agents that can be exploited and always try to appease other parties are not viable.
- The generalization results show that the model basically learns a static "cooperate/defect" policy and applies this to the different games. It would be much more interesting to see whether LLMs can adapt and behave differently depending on different payoff structures and strategic situations.
- Currently I do not see where the "LLM" really comes into play, other than being able to generalize to descriptions of other bimatrix games (though see my concerns about generalization above). My current impression from the results is that they don't really go beyond results one could have achieved with a simple parameterized conditional policy with 6 real-valued parameters. This is also supported by Section 8.9 in the appendix. Though I might have misunderstood something.
- Overall, studying training in games and using potential non-game rewards to incentivize cooperative equilibria is an interesting direction. However, I believe the experiments in the paper at hand are unable to make useful progress on this.

**Questions:**

- Do the models really understand the strategic situations they are in? One point of evidence in favor is that they seem to be able to find the equivalent "cooperate" and "defect" actions in other games, but it would be nice to have more evidence, such as asking the models follow-up questions about the games. Do you have any other evidence whether the models understand their strategic situation?
- I believe the background section is a bit too long. There is a lot of generic text that is not necessary to understand the core experiments with the ipd.
- Instead, I would be interested to see some examples of the games in the main text.
- From looking at Figure 6, I believe LLMs will be able to pattern match the payoffs to a prisoner's dilemma and thus infer that this is about prisoner's dilemmas. It would be interesting to use actual paraphrases of strategic situations which are ipd-like but don't have an explicit payoff matrix.
- I wouldn't use the term "unlearning" for the thing studied in this paper. Unlearning is a technical term used to describe removing knowledge from a model, not teaching the model a new strategy. The thing studied here is simply fine-tuning with a different initialization.
- I find the font in the plots for Figure 2 a bit small and hard to read.
- Is there a reason you use memory 1 policies? In principle, more than one iteration of the game would fit into an LLM's context window and would allow the LLM to make more sophisticated strategy judgments.
- Is it right that LLMs basically generalize by identifying the "cooperate" and "defect" options and then using the same policy as in the ipd? Have you tried permuting the two action labels and the columns/rows of the game descriptions? Have you observed cases where models change their actions in accordance with correct deontological judgments?
- Line 395+396 “static opponent” - is this still the tit for tat opponent?

---

> ### Author Response · Authors · 2024-11-25
> **Response to Reviewer Jn1h (1/6)**
>
> We thank **Reviewer Jn1h** for their detailed feedback and for their extensive questions. Please find our answers below:
>
> ### Question 1:
> > “Do the models really understand the strategic situations they are in? One point of evidence in favor is that they seem to be able to find the equivalent "cooperate" and "defect" actions in other games, but it would be nice to have more evidence, such as asking the models follow-up questions about the games.”
>
> The question of understanding is certainly relevant and interesting, though hard to address fully, as “understanding” in LLMs is still a hotly debated topic (see, e.g., Mitchell & Krakauer, 2023; Shanahan et al., 2023, Shanahan, 2024). We personally take a somewhat “functionalist” view of these models, given the next-token-prediction at the core of their function. While instruction-tuning or alignment-tuning may make these models behave in a more goal-directed way, we are wary of anthropomorphizing them with terms like “understanding”. Even if the models can correctly predict the next tokens in response to a follow-up “understanding” question, this would not necessarily demonstrate that the model truly understood its environment or the decision-making process. Therefore, in our core analyses in the paper, we focus on the available evidence for strategic decision-making capabilities of these models, rather than an intrinsic “understanding”.
>
> Following the suggestion of **Reviewer Jn1h**, we also conducted an analysis where we asked a sample of models follow-up questions about their decisions. In particular, we asked two types of questions (in independent runs): “Why did you make this decision?” and “What strategy did you use to make this decision?”. The format of the prompt is presented in Figure 26, and sample results are presented in Figure 27. We find that the Gemma models often come up with plausible explanations in response to the explicit question about “strategy”, but are less coherent in response to the simple question of “why they made this decision” (see Section 8.12 of the Appendix for further discussion). In the case of both questions, the explanations are not always correct and do suggest some matching to example explanations seen in training data, without “reasoning” about the current situation or state correctly.
>
> Returning to the decision-making scenarios, the analysis of permutations of the order or presentation of each action pair in the game prompt is relevant to this question. We agree with **Reviewer Jn1h** that our models exhibit some generalization capabilities in the strategic decision-making process at test-time. Specifically, they *“seem to be able to find the equivalent "cooperate" and "defect" actions in other games”*, especially after we swap the action symbols from “action1” and “action2” to “action3” and “action4” at the evaluation step. Following the Reviewer’s suggestion, we performed a stress-test of the models’ learnt policies by testing a wider range of permutations of the ordering of actions in the payoff matrix. The prompts and results are presented in Figures 9, 22 and 23 in the paper. We discuss the results in our response to the Reviewer’s specific question about permutations below.
>
> References:
> - Mitchell, M. and Krakauer, D.C. 2023. The debate over understanding in AI’s large language models, PNAS. 120 (13) e2215907120.
> - Shanahan, M, et al. 2023. “Role Play with Large Language Models.” Nature, vol. 623, no. 7987, pp. 493–98.
> - Shanahan, M. 2024. “Talking about Large Language Models.” Communications of the ACM, vol. 67, pp. 68–79.
>
> ### Question 2:
> > “I believe the background section is a bit too long. There is a lot of generic text that is not necessary to understand the core experiments with the ipd.”
>
> We included a discussion on LLMs as agents, along with background information on moral alignment and fine-tuning, to position our paper within the AI Alignment literature and provide sufficient context for a general audience. In our work we aimed to make a core methodological contribution to the area of AI Alignment rather than purely to the specific decision-making / IPD domain, therefore we believe the setting of this context in the background section is important.
>
> ### Question 3:
> > “Instead, I would be interested to see some examples of the games in the main text.”
>
>
> Our paper introduces the games in Section 2.3, presenting the description of the core elements of the dilemma game as well as the payoff matrix for the IPD. We have now added a sentence giving an example of a real-life IPD game in lines 180-182, involving two housemates cleaning a house. In our original draft, we had also included the full prompt in Figure 6 of the Appendix - we have now added a reference to that figure in line 25 of the main text. Having said that, we would be happy to add additional information if these modifications are considered not sufficient. We would like to ask the Reviewer if they had any further potential examples in mind.

---

> ### Author Response · Authors · 2024-11-25
> **Response to Reviewer Jn1h (2/6)**
>
> ### Question 4:
> > “From looking at Figure 6, I believe LLMs will be able to pattern match the payoffs to a prisoner's dilemma and thus infer that this is about prisoner's dilemmas. It would be interesting to use actual paraphrases of strategic situations which are ipd-like but don't have an explicit payoff matrix.”
>
> We agree with the general concern of pattern-matching to the IPD. However, studying fine-tuning systematically using unstructured or paraphrased prompts would have been very difficult since the choice of particular situation or wording may influence results. In fact, in our early experiments pre-fine-tuning we tested out both the structured and unstructured IPD prompts, and found that the model’s choices seemed less biased towards default cooperation in response to the abstract, structured prompt, which is why we chose it for fine-tuning. We also made a conscious choice to use the “neutral” action tokens “action1” and “action2” in fine-tuning to make our approach more systematic, rigorous and clean, without inventing extra variables such as scenario specifics or wording.
>
> To test potential prior knowledge of the IPD, in our original version of the paper, we also conducted test-time evaluations with an explicit IPD prompt (see bottom prompt in Figure 10), where payoffs needed to be assumed by the model. Following the above suggestion of **Reviewer Jn1h**, we have now conducted two further evaluations using:
>
> - an unstructured IPD prompt (no payoff matrix, numeric payoffs described in text);
> - a paraphrased IPD-like situation prompt (no payoffs at all; action tokens associated with specific examples: “action3” = “clean the house”; “action4” = “wait for your flatmate to clean the house”).
>
> The four different IPD-related prompts are now presented in Figure 10. We analyze the action types (i.e., action | state) of each model in response to these in Figure 17. We consider this a preliminary evaluation, which might be the topic of an entire project/paper focusing extensively on this interesting analysis.
>
> The results show that the paraphrased IPD-like prompt was more effective for the base model, generating responses with legal action tokens (see Figure 17, left). It is possible that this paraphrased prompt, reflecting the situation in plain language, was itself pattern-matched to the model’s training data more closely than the abstract, structured format used in our fine-tuning. Specifically, real-life examples are often used to describe the IPD in textbooks, so the model may pattern-match a paraphrased scenario just as easily as a prompt containing a payoff matrix.
>
> Our results for fine-tuned models suggest that these were able to generalize their moral policies reasonably well from the structured training prompt to the unstructured IPD prompt, as action choices are very similar between these two prompts (notably, despite our use of new action tokens “action3” and “action4” at test time). However, as we move onto prompts that did not contain a payoff structure (“IPD-like situation” and “Explicit IPD”), action choices become closer to random, though still leaning on defection by the agent fine-tuned on game payoffs, and leaning on cooperation by the agents fine-tuned on Deontological or Utilitarian rewards.
>
> As a preliminary test of further pattern-matching beyond dilemma games, we would like to highlight an element of our analysis from the original draft of the paper. We analyzed the impact of IPD-based fine-tuning on responses to other, non-matrix-game prompts containing potentially recognizable elements such as an action choice, a “state”, and/or a “game” description, but no payoffs at all (see Figure 11). The results (see Figure 18) suggest that fine-tuning on the IPD made the models more likely to learn an order preference over the two action tokens available in the prompt, even when these tokens do not have any cooperative or defective associations in the prompt itself. For example, the earlier “action3” token is preferred over “action4” by the prosocial models, similar to their preference of “action1” (=”cooperate”) over “action2” (=”defect”) during training.
>
> While we conducted this extra analysis to satisfy the Reviewer’s (and our) curiosity, we would like to stress again that in this work, we specifically focus on generalization to situations which require a payoff matrix. In other words, we examine games where outcomes are quantifiable. This relies on a body of fundamental literature in economic games designed to analyze a variety of real-life situations, as highlighted by **Reviewer kTWD**. In order to more extensively and systematically study generalization beyond games with payoffs, we would need a different setup to train agents on many different games described using text - though note the discussion on systematicity, phrasing and situation choice discussed above. Such a setup lies beyond the scope of this study.

---

> ### Author Response · Authors · 2024-11-25
> **Response to Reviewer Jn1h (3/6)**
>
> ### Question 5:
>  > “I wouldn't use the term "unlearning" for the thing studied in this paper. Unlearning is a technical term used to describe removing knowledge from a model, not teaching the model a new strategy. The thing studied here is simply fine-tuning with a different initialization.”
>
>
> We agree that the term unlearning can be used to describe a particular technical approach to deleting knowledge from a model, which differs from the one that we take in this paper. In the context of our paper, we qualify the term to mean unlearning a selfish strategy - i.e., re-prioritizing certain previously developed principles for decision-making. We have added a footnote in line 71 to clarify the difference, and also made this use of the term more prominent in labeling Figure 3, as suggested by *Reviewer kTWD*.
>
>
> ### Question 6:
> > “I find the font in the plots for Figure 2 a bit small and hard to read.”
>
>
> We thank the Reviewer for flagging this issue. We have now re-plotted Figures 2 and 3 to place the legend outside the chart and with an increased font size. We hope that this addresses the Reviewer’s concern, but we are happy to revise the figures again if this is not the case.
>
>
> ### Question 7:
> > “Is there a reason you use memory 1 policies? In principle, more than one iteration of the game would fit into an LLM's context window and would allow the LLM to make more sophisticated strategy judgments”
>
> We agree that using longer history lengths might allow us to observe more interesting strategic behaviors. However, in practical terms, if we go beyond a memory length equal to 1, interpreting the resulting strategies (action | state) becomes much harder. Ease of interpretation is the reason why an analysis based on history length equal to 1 is often found in other literature in the field, and why we have decided to use history length of 1 in this instance. Furthermore, using the history length of 1 allows for our methodology to apply to a broader set of real-world models, including environments with a very large number of agents, where any pair of agents only interact rarely with one another (without repeated encounters beyond one recent experience).

---

> ### Author Response · Authors · 2024-11-25
> **Response to Reviewer Jn1h (4/6)**
>
> ### Question 8:
> > “Is it right that LLMs basically generalize by identifying the "cooperate" and "defect" options and then using the same policy as in the ipd? Have you tried permuting the two action labels and the columns/rows of the game descriptions? Have you observed cases where models change their actions in accordance with correct deontological judgments?”
>
> In the original draft we analyzed the impact of two permutations of the order of the rows and columns in the payoff matrix (the top and bottom prompts in Figure 9 of the Appendix). Following the Reviewer’s suggestion, we now present a more complete analysis of all 4 possible permutations of the payoff matrix - please see Figure 9 for the prompts, and Figures 22 and 23 for the results. We find that models fine-tuned on Deontological and Utilitarian rewards are generally able to generalize their learned policies to the first three permutations. However, for agents fine-tune on game payoffs, we observe that permuting the order of the rows (i.e., an agent’s own actions) in permutations 3 and 4 confuses the player more than permuting the order of the columns. In the case of fully reversed rows and columns - the prompt most distant from the one used in training (permutation 4), selfish agents fine-tuned with game rewards appear more cooperative than the morally fine-tuned Utilitarian and Deontological agents. As shown in Figure 23, this unlikely behavior can be explained by the fact that the players actually just choose the latter of the tokens being presented in the matrix, which in this case is “action3”. Given the fact we used new action tokens at test time, this again suggests that the models might have learned to ascribe some sort of meaning to the relative order of the two action tokens presented in the test prompt (e.g., select the first of the two action tokens as it tends to indicate cooperation), and this relationship breaks if we present the payoff matrix in reverse order at test time. Analysing the reason why Deontological players’ learnt policies are especially robust across permutations (both in terms of regret - Figure 22, and actions taken - Figure 23), we find that this because Deontological players are able to keep a large amount of their actions cooperative despite the change in ordering, and they are still able to confirm to their norm of not defecting against a cooperator even in permutation 4 (see Figure 23). Models fine-tuned on Utilitarian reward, on the other hand, start defecting 50% of the time on permutation 4 (i.e., the prompt where both rows and columns are reversed). This might point to slightly better generalization through the Deontological fine-tuning than through the Utilitarian fine-tuning.
>
> As mentioned in our original draft, we also performed a test-time evaluation using the original action tokens “action1” and “action2”, but with their meaning reversed in the prompt (see prompt in Figure 8b and results in Figure 21). As discussed, we found that swapping the meaning of action tokens at test time resulted in weaker performance in terms of both Deontological and Utilitarian regret. This shows that indeed the fine-tuned models also learned to attach a meaning to the specific action tokens used in training (specifically, they learned to associate “action1” with cooperation, “action2” with defection), and they were unable to let go of this meaning completely at test time, even when the prompt required for them to do so.
>
> ### Question 9:
> > “Line 395+396 “static opponent” - is this still the tit for tat opponent?
>
> Yes, thank you - we have added a clarification in brackets.

---

> ### Author Response · Authors · 2024-11-25
> **Response to Reviewer Jn1h (5/6)**
>
> ## Comments on strengths & weaknesses:
> We thank **Reviewer Jn1h** for complimenting our approach of developing new alignment techniques via intrinsic reward design without requiring human feedback. We also thank the Reviewer for complementing our presentation and soundness.
>
> ### Comment 1:
> We agree with the Reviewer that **training on purely-deontological or purely-utilitarian rewards** is only a starting point, and developing approaches for fine-tuning LLM agents with multi-objective rewards is of great interest. As a first step in this direction, in our original draft we evaluated the actions of an agent trained not only with Deontological rewards, but also using Game + Deontological rewards. Game + Utilitarian rewards are harder to combine in a meaningful, linear fashion, since they directly contradict each other - either a player optimizes for their own game payoff, or for the sum of the two players’ payoffs.
>
> LLM systems and agents can arguably already address the needs of the user (i.e., “play the game” following the game-theoretic principle of reward maximization) quite well. In this study we seek to advance the alignment and safety of these systems in particular, which is an additional goal on top of successfully executing the task. To reiterate, aligned behavior is the core goal of this study. A first essential step towards developing a methodology for aligning agentic behavior is to study straight-forward, prosocial reward functions in a traditional prisoner's dilemma environment used to model real-world conflict scenarios. Future work can indeed extend this methodology to further multi-objective implementations, beyond the Game+Deontological reward case study presented in this work.
>
> We would like to take this opportunity to comment on the idea of ‘hand-designed’ rewards brought up by the Reviewer. Indeed, in our paper we provide an example of a reward structure for an environment with two possible actions. We would argue that designing (“hand-crafting”) rewards is an essential part of RL. The rewards in the Utilitarian agent are based on the original payoffs of the game, but these payoffs have themselves been hand-crafted to describe the goals and outcomes of a situation in the original game. In our paper, we show that it is possible to quantify moral decision-making by designing custom moral reward functions. The general methodology underlying our approach - for example, the quantification of the Deontological principle of “do not be mean to nice people” - can be generalized to other environments and action spaces.
>
> Finally, we would also like to highlight the difference between the Deontological agent and TFT: it is true that both of these agents cooperate against an opponent who previously cooperated - a sort of reciprocal morality; however, when facing a defector, the Deontological agent can either cooperate or defect (as shown in Figure 5 - they choose either action with equal probability across the five games), whereas a TFT agent would necessarily defect against a defector.
>
>
> ### Comment 2:
> > “It would be much more interesting to see whether LLMs can adapt and behave differently depending on different payoff structures and strategic situations.”
>
> We would like to highlight that in the paper we do explicitly test 4 other matrix game scenarios in which different strategies are required vs the original IPD game (Figure 4). In particular, one of these is a game of our own design, in which cooperation is specifically not the optimal policy (see Iterated Defective Coordination). By analyzing Utilitarian moral regret on this game in particular, we show the specific weaknesses of fine-tuning with Utilitarian rewards in particular: models fine-tuned with this reward on the IPD do indeed show a bias for cooperation, which they are unable to overcome at test-time in a game where defection would have been undeniably better in terms of their moral reward function.
>
> In terms of generalization to other strategic situations which require an action choice without well-defined payoffs, we point **Reviewer Jn1h** to our analysis of responses to 3 general call-to-action prompts (see Figure 11), with results presented in Figure 18 and discussed above.
>
> Finally, in Figure 19 (and Section 8.8), we specifically analyze the generalization of learned reciprocity (i.e., the Tit for Tat principle, or “do to others as they have done to you in the past”) to scenarios that do not involve a payoff matrix - specifically, the “Action+Game+State” prompt from Figure 11, and in which therefore no mapping to known “cooperate” or “defect” strategies is possible. We show that reciprocity learned on the IPD (i.e., cooperate against cooperators, defect against defectors) is not generalized to this abstract “Action+Game+State” situation without a payoff matrix, as reciprocated actions are much more rare than in response to the original structured IPD prompt.

---

> ### Author Response · Authors · 2024-11-25
> **Response to Reviewer Jn1h (6/6)**
>
> ### Comment 3:
> > “Currently I do not see where the "LLM" really comes into play, other than being able to generalize to descriptions of other bimatrix games (though see my concerns about generalization above). My current impression from the results is that they don't really go beyond results one could have achieved with a simple parameterized conditional policy with 6 real-valued parameters. This is also supported by Section 8.9 in the appendix. Though I might have misunderstood something.”
>
> We thank the Reviewer for the comment and would like to offer a clarification. The main purpose of this particular work was not to build the strongest game player (given the classic social dilemma pay-offs), as in most of the existing literature on the topic. Instead, we aim to advance methods in aligning an LLM-based system. We focus on developing a method for moral alignment, looking to modify the behavior of a particular class of systems that already underlie some decision-making agents today. We base our methodology on past research in Game Theory and Multi-Agent Reinforcement Learning (see also the answer below), but the core aim of the work is to develop methods for moral alignment of LLMs in particular.
>
> ### Comment 4:
> > “ studying training in games and using potential non-game rewards to incentivize cooperative equilibria is an interesting direction.”
> While we agree that this is an interesting direction, we would like to once again highlight that the goal of this study was not to contribute further advances to general strategic decision-making in learning agents. In this work, we especially focused on applying the concept of intrinsic rewards to fine-tuning real-world agentic systems which are emerging today, namely LLM agents. This relates to our answer above about the applied nature of this study and our focus on alignment.
>
> Advancing the understanding of designing sophisticated intrinsic rewards for training general agents to play games more cooperatively is indeed a fascinating direction, but a different one from the one we aim for here. For examples of foundational works in this direction, we refer the interested reader to Hughes et al. (2018); McKee et al. (2020).
>
> References
> - Hughes, E. et al. 2018. “Inequity aversion improves cooperation in intertemporal social dilemmas.” NeurIPS’2018.
> - McKee, K. et al. 2020.. Social diversity and social preferences in mixed-motive reinforcement learning. AAMAS’2020, pp. 869–877.

---

### Official Review · Reviewer_kTWD · 2024-11-05

**Soundness:** 4
**Presentation:** 4
**Contribution:** 4
**Rating:** 8
**Confidence:** 3

**Summary:**

This paper points out a fundamental limitation of most current alignment approaches, which is that preference data is obtained from human participants that are highly noisy expressions of underlying values and end up in opaque models that are not interpretable.  The authors point out that a good alternative to this would be designing reward functions that explicitly encode values (or patterns of moral behavior) and use them for Reinforcement Learning-based fine-tuning of foundation agent models.  They validate their approach by constructing an iterated prisoner's dilemma environment and seeing how their fine-tuned models (tuned with various value-based policies) respond to tit-for-tat players or other LLM participants.

**Strengths:**

This is a fantastic paper!  It addresses a critical and highly understudied problem with a clever and interesting solution, the experiments are clear, simple, and well-executed and the writing is direct and easy to follow.

**Weaknesses:**

Though I love what the authors are doing here, a potential weakness of this approach, is encapsulated by this sentence in the introduction: "In
theory, our solution can be applied to any situation in which one can define a payoff matrix that captures the choices available to an agent that have moral implications."  But of course the trouble is that the world is very messy and complex place that often cannot be easily captured in a payoff matrix.  Now, the reason that economic games have gotten so much traction in the behavioral sciences is that they nicely capture the dynamics of the social world in neat packages that can be quantitatively studied in highly controlled environments -- and indeed the authors capitalize on that advantage.  Can they at least gesture at the challenges (and potential solutions?) to bridge their strategy with more open-world contexts?

**Questions:**

Do the authors think that an interesting comparison to their system (fine-tuning via Reinforcement Learning with Intrinsic Rewards) would be simply prompting LLMs (eg GPT) do follow a natural language directive equivalent to the moral values they tested (eg, Deontological, Utilitarian)?  On the one hand, I could see this being an interesting baseline to beat.  On the other, there is much to be gained from the system that this paper describes beyond just the performance on these tasks, so I could see this comparison feeling a bit silly.  Curious what the authors think about this.  In general, it feels like having some baseline to compare performance to would be a good idea.


Small points:
Fig 3: label more prominently as depictions of the "unlearning" experiment.

---

> ### Comment · Reviewer_Jn1h · 2024-11-19
>
> I agree designing desired moral principles explicitly rather than using implicit preference comparisons is a good direction. There is work on this under the heading of constitutional AI or model specs. I worry that the approach taken in this paper is too simplistic, with hand-crafted rewards and no signs of nontrivial generalization, so that it doesn't really contribute relevantly to this important direction.

---

> > ### Author Response · Authors · 2024-11-25
> > **Response to additional comment by Reviewer Jn1h on the review of Reviewer kTWD**
> >
> > We thank the reviewer for pointing out these two existing methodologies. One of these (Constitutional AI) was already mentioned in our Discussion section, where we suggested a way of integrating our morally fine-tuned agents into a "constitution" for training other models with AI feedback. We have now added a more extensive discussion about the differences between our solution and these approaches in the "Summary of the responses and changes" (see sub-heading "Comparison against existing implementations").

---

> ### Author Response · Authors · 2024-11-25
> **Response to Reviewer kTWD**
>
> We thank **Reviewer kTWD** for their enthusiastic comments, kind feedback and their questions.
>
> 1. We agree with the Reviewer’s suggestion of *“simply prompting LLMs (eg GPT) to follow a natural language directive equivalent to the moral values they tested (eg, Deontological, Utilitarian)”* as an interesting additional baseline. An example real-world implementation of this is Model Specs for GPT (OpenAI, 2024), as mentioned by **Reviewer Jn1h**. Simply prompting the models to adhere to certain values explicitly could prove effective, provided that the underlying LLM has some “commonsense” representation of these values from its pre-training. However, the model would still need to take the “reasoning” step of translating from the word description of a value to the specific action tokens, strategies and outcomes available in these particular games. While larger models like GPT may be capable of taking such a “reasoning” step, it remains uncertain whether smaller models, such as Gemma, can do so effectively.
>
>     To potentially test this, following **Reviewer kTWD**’s suggestion, in the time available (and within the computational constraints of our institution), we tried to analyze the value-prompting baseline as an extra test, to compare with no fine-tuning at all and the rewards-based fine-tuning we present in the paper. We prompted the original Gemma model with a simple description of a Utilitarian or Deontological value (see Figure 24). With these two new baselines, we evaluated action choices and regret across the same five games. The results of this analysis are shown in Figure 25. Unfortunately, as we saw with the base non-fine-tuned model, the vast majority of the responses from these value-prompted models contained illegal tokens. Thus, we were unable to see the impact of value prompting on potential model behavior. One way we could address this in the future is to fine-tune each model on legal tokens first via supervised learning, before performing any value-specific fine-tuning or baselining.
>
>
> 2. We would also like to address the request by **Reviewer kTDW** to *“at least gesture at the challenges (and potential solutions?) to bridge [our] strategy with more open-world contexts?”*
> We agree with the Reviewer’s description of the relative pros and cons of economic games as models of real-world situations, and also believe that it would be of interest to extend moral fine-tuning to a broader set of situations than structured matrix games.
>
>     One way to extend the generality of our approach beyond the structured game scenarios is to use more open phrasing in the prompt. In our original draft, we made some efforts to address this by analyzing the responses of models to other abstract prompts involving action choices and states, but no payoffs (see prompts in Figure 11, results in Figure 18 - now slightly modified for clarity).
>
>     In response to **Reviewer kTDW**’s question and **Reviewer Jn1h**’s suggestion, we have now further assessed generalization to four different IPD-like prompts, including an unstructured IPD prompt and a prompt describing an IPD-like everyday situation without any explicit payoffs (see Figure 10). This new analysis tests whether models behave consistently in situations that have similar strategic decision-making structures and action tokens to the IPD training prompt, but which are described in different wording, more implicitly or more explicitly. The results of this analysis (Section 8.10 and Figure 17) show that fine-tuned agents play similar strategies across variations of the IPD prompt at test-time, but the impact of their moral fine-tuning decreases slightly as we move further from the original structured prompt - for example, Utilitarian-tuned agents start defecting slightly more on the unstructured IPD prompt, and even more on the paraphrased or explicit IPD prompts.
>
>     Beyond these environments, one might compare our implementation with existing alignment techniques in general open-ended language production settings. Prominent approaches for defining explicit principles for model behavior include Model Specs (OpenAI, 2024) and Constitutional AI (Bai et al., 2022), as brought up by **Reviewer Jn1h**. As we suggest in our paper’s Discussion section, our moral agents can be used as part of a “constitution” to provide feedback to other models. The challenge in this approach would be to first validate whether the moral policies learned by agents acting in an environment allow them to provide valid moral feedback to other decision-makers - this is not a given. To bridge these methods with our focus on decision-making, a novel approach would be to use a “constitution” of such models to provide feedback on decision-making scenarios for LLMs in particular.
>
> References:
> - Bai et al., 2022. Constitutional AI: Harmlessness from AI feedback. arXiv Preprint arXiv:2212.08073.
> - OpenAI. 2024. Model Specs. https://cdn.openai.com/spec/model-spec-2024-05-08.html

---

### Author Response · Authors · 2024-11-25
**Summary of the Responses and Changes (2/2)**

## Comparison against existing implementations:
In response to **Reviewer Jn1h**’s comment, we would like to clarify the difference between our method and existing implementations such as Constitutional AI (Bai et al., 2022) and Model Specs (OpenAI, 2024).

*Constitutional AI*:
- Constitutional AI was essentially developed for generic NLP applications, such as conversational systems. As discussed in the paper, we aim instead to develop a methodology specifically for strategic decision-making agents based on LLMs.
- Our approach and Constitutional AI share the idea of representing explicit values as learning signals for RL-based fine-tuning. This explicit nature allows for interpretability of the overall underlying value (reward) system adopted by the system designers, as compared to traditional RLHF. The proposed solution does not rely on feedback from very large models and, therefore, it is also very efficient from a computational point of view. In fact, the fine-tuning we performed was done on a single GPU and over a short time period, making our approach potentially cheaper and more scalable to one’s individual preferences than methods such as Constitutional AI.
- That said, as we mention in our paper, our method can be combined with a “constitutional” approach as follows: “Finally, agents trained via intrinsic rewards as proposed in this study could also form the basis for a Constitutional AI architecture composed of artificial agents characterized by different moral frameworks (Bai et al., 2022).”

*Model Specs*:
- Model Specs, similar to Constitutional AI, express explicit principles across a range of domains in plain language. The Model Specs published for GPT, for example, express complex ideas, such as agent objectives and legal obligations, in plain text - for example by specifying rules. There is an implicit expectation that the models can then meaningfully reason from the textual principle to aligned responses. While this may have proven somewhat effective for conversational agents, it is not clear how well these principles translate to agentic uses of LLMs, where models take actions in an environment.
- Model Specs are currently used for prompting (post-training) rather than fine-tuning models. We instead study fine-tuning in this work as a more fundamental way of embedding moral strategies in LLM-based agents.
- Nevertheless, as suggested by **Reviewer kTWD**, we have now tested a somewhat similar prompting approach in our domain - though of course at a smaller scale - by adding two potential baselines of models pre-prompted to behave according to certain values (see Figures 24 and 25). Unfortunately, in our implementation, this did not allow us to meaningfully steer model behavior, as models simply output illegal tokens in response to these prompts, similar to the base model we tested in the original results. We believe that the idea of specifying principles in text for more advanced models is indeed interesting, and in future work we plan to test the validity of this approach for moral reasoning in particular.

References:
- Bai et al., 2022. Constitutional AI: Harmlessness from AI feedback. arXiv Preprint arXiv:2212.08073.
- OpenAI. 2024. Model Specs. https://cdn.openai.com/spec/model-spec-2024-05-08.html

---

### Author Response · Authors · 2024-11-25
**Summary of the Responses and Changes (1/2)**

We thank all the Reviewers for their insight and suggestions. Below we present a summary of the changes made and responses to key points raised.

## New results added:
- As suggested by **Reviewer Jn1h**, and to begin to address the question of **Reviewer kTDW**, we test the generalization of fine-tuned models to IPD-like prompts in other, potentially more open-ended formats. We now test 4 IPD-like prompts in total, adding two new ones (see prompts in Figure 10 and results in Figure 17 and Section 8.10):
        - [old] structured IPD prompt with a payoff matrix - identical to the training prompt
        - [new] unstructured IPD prompt with textual descriptions of the payoffs
        - [new] paraphrased IPD-like scenario using a real-life example and no payoffs at all
        - [old] explicit IPD prompt with payoffs assumed based on past knowledge of the game
- As suggested by **Reviewer kTWD**, we test 2 additional baselines: models prompted for the Deontological or Utilitarian moral value (see Figures 24 for prompts, Figure 25 for results, Section 8.11 for the discussion).
- As suggested by **Reviewer Jn1h**, we now test 4 permutations of row & column ordering in the payoff matrix within the game prompt (see Figure 9 - previously we only had two of these). The results of these permutations are presented in Figures 22 and 23.
- As suggested by **Reviewer Jn1h**, we ask the fine-tuned models follow-up questions about the strategic situations. In particular, we ask a sample of fine-tuned and base models the following follow-up questions: 1) Why did you make this decision?; 2) What strategy did you use to make this decision? Figure 26 represents the prompt format, and example responses are presented in Figure 27. Descriptive text has been added in Section 8.12.

## Changes made to the draft of the paper:
 - We split out Figure 8 into Figures 8 and 9 (adding new permutations as suggested by **Reviewer Jn1h**).
- We modified Figure 10 to include two extra prompts, as suggested by **Reviewer Jn1h** - these were used to analyze the generalization of fine-tuned models to IPD-like prompts in different formats; added Figure 17 to visualize the results of this analysis.
- We moved the results for explicit_IPD prompts from Figure 18 to Figure 17, to make the comparison with results for other IPD-like prompts (see line above) easier.
- In Figures 18, 19, 20 and 21: removed visualizations for training vs an LLM opponent, as there are now too many charts in the Appendix for the reader to process, and these do not contribute core insights.
- Added Figures 22, 23, 24, 25, 26, 27 and text within the Appendix to describe the analyses requested by **Reviewer Jn1h** and **Reviewer kTWD**. We believe that these analyses also partially address the concerns of **Reviewer yMDj**.
- Modified all plots in the Appendix to represent analyses with new action tokens action3 and action4 (unless otherwise stated) - for consistency with the results reported in the main text of the paper and the newly added analyses.
- Small edits to figure labels, font sizes and phrasing to address the Reviewers’ specific requests (Font size increased in the legend of Figures 3 and 4, as requested by **Reviewer Jn1h**; font label clarified in Figure 4, as requested by **Reviewer kTWD8**; text clarified and an example of an IPD-like situation added as requested by **Reviewer Jn1h**).
- Added clearer headers to all Appendix figures to guide the reader through the increased amount of results.

## Note on extension to more complex scenarios:
- As we note in the paper, we agree that an extension of this method to more complex scenarios would be of great interest, and it is indeed part of our research agenda. We believe that the simple, widely-used 2x2 matrix games provide a very interesting, insightful, systematic and interpretable case study. It is worth noting that many initial works on Reinforcement Learning based decision-making agents are indeed based on social dilemmas and similar games. To the best of our knowledge, this is the first paper to perform LLM agent fine-tuning with intrinsic rewards defined for these scenarios: for this reason, we prioritized depth of analysis over breadth at this stage.

---

> ### Comment · Reviewer_Jn1h · 2024-11-26
>
> Thank you for these comprehensive additional analyses and for your in-depth response to my review. I now believe I understand better what is going on based on all the experiments in the paper. My overall impression of the results is that they are somewhat negative in the sense that generalization to different prompt formats and different strategic situations is not very robust. I do think the inclusion of these results makes the paper more valuable to the community and I am grateful for the authors' efforts here.

---

> > ### Author Response · Authors · 2024-11-27
> > **Response to the additional comment from Reviewer Jn1h**
> >
> > We thank **Reviewer Jn1h** for the positive reply, their helpful suggestions earlier in the rebuttal process, for highlighting how the extra analysis and clearer presentation of our results help make the paper more valuable for the community, and for raising the score.
> >
> > We are happy to seek to address the remaining concerns of the reviewer regarding our results. The primary goal of this study was to develop moral fine-tuning that works for a specific game/payoff structure - in our case, the IPD. For this environment in particular, we show that the fine-tuning is robust, working effectively for six different reward definitions (see Figures 2 and 3), against five types of opponents (Figure 14), and even across 3 other phrasings of the game (Figure 17). We elected to also stress-test and explore limit cases of our methodology. We believe it is important to show where the approach can be further improved, given the probabilistic nature of the underlying models and the associated fine-tuning process. As part of this stress-testing, we have explored a constrained form of generalization by looking at games with other payoff structures, and found both strengths and weaknesses of particular moral reward functions (Figures 4a and 5). As we discussed throughout the rebuttal, more global generalization would require training across a range of games from the beginning. We agree that this is indeed an exciting direction for future work.

---

### Meta-Review · Area_Chair_vLJq · 2024-12-20

**Metareview:**

This paper was about evaluating LLMs using deontological and utilitarian ethical frameworks and fine tuning using hand-designed rewards to reduce selfish behavior. There was agreement among the reviewers that the problem addressed by this paper was novel and important, and that the experimental evaluations were especially interesting.

**Additional Comments On Reviewer Discussion:**

One reviewer was worried about the hand-tuned nature of the proposed fune tuning method but they stayed quite involved in the discussion and ultimately came around to the view that their initial concern, while still present and potentially reducing generality, should not be seen as an impediment to accepting the paper, given that it was also interesting for other reasons having to do with the evaluation methods.

---

### Decision · Program_Chairs · 2025-01-22

Accept (Poster)